# A Multi-Criteria Decision-Making Technique Using Remote Sensors to Evaluate the Potential of Groundwater in the Arid Zone Basin of the Atacama Desert

Víctor Pocco [1], Samuel Chucuya [2], Germán Huayna [2], Eusebio Ingol-Blanco [3] and Edwin Pino-Vargas [2,*]

[1]  Department of Geology-Geotechnics, Jorge Basadre Grohmann National University, Tacna 23000, Peru
[2]  Department of Civil Engineering, Jorge Basadre Grohmann National University, Tacna 23000, Peru
[3]  Department of Water Resources, Universidad Nacional Agraria La Molina, Lima 15024, Peru
*  Correspondence: epinov@unjbg.edu.pe; Tel.: +51-952298638

**Abstract:** One of the most notable problems in the Atacama desert is the low occurrence of rainfall, which leads to a shortage of surface and groundwater for different users in the region. Therefore, the task of carrying out new exploration studies of potential groundwater zones (GWPZs) is of vital importance for decision-makers in water resources. The main objective of this research is to determine potential sources of groundwater using a Multi-Criteria Decision-Making technique with remote sensors. A method of exploration using the Analytical Hierarchy Process (AHP) techniques applied to remote sensing data is provided. The AHP method allows calculating the influence of multiple factors, and along with the GIS environment, a map of groundwater exploitation potential can be produced. The results indicating GWPZs showed four classifications of groundwater potential. The distribution shows 15.02%, 23.93%, 59.80%, and 1.25% of the total area with high, moderate, low, and very low potential, respectively. The results were validated with existing wells in the study area, offering an acceptance of 86.9%. This reveals the effectiveness and accuracy of the AHP-based GIS approach as a strategy for analyzing groundwater potential in arid zones. Similarly, the tested high GWP areas are helpful for the development and management of water resources in the Caplina basin.

**Keywords:** groundwater potential; water shortage; AHP; multi-criteria decision; remote sensors; arid region

## 1. Introduction

The impact of climate uncertainty is already evident in semi-arid and arid regions, facing increased vulnerability to water scarcity [1,2]. A dry climate and population growth increase the demand on scarce water resources [3]. The Atacama Desert in northern Chile and southern Peru is a hyper-arid region stretching almost 1000 km in South America (latitudes 19° S to 30° S) and is considered the oldest and driest desert on Earth [4–6]. Its hyperacidity is subject to various factors, such as the subtropical subsidence of the planet [7], the rain-shadow effect caused by the Andes mountain range [4], and the precipitation inhibition effect of the Humboldt Current [8].

The overexploitation of authorized and unlicensed wells for irrigation and consumption affects groundwater, causing qualitative and quantitative degradation [9,10]. Seawater intrusion is another of the main problems in groundwater quality management [11]. Similarly, due to governance conflicts, laws regarding groundwater resource management and limitation regimes have been altered [12].

In this sense, identifying potential groundwater zones still presents a challenge in terms of time and cost to resolve the problems of scarcity and management of these systems [13]. Similarly, government-managed projects to increase water supply tend to take a long time due to poor administrative management [12,14]. Thus, the precise delimitation of Groundwater Potential Zones (GWPZ) is essential for the sustainable management of

water resources in arid areas [15,16]. In this way, geospatial technology, such as the use of satellite data can provide a scientific analysis contribution from a general viewpoint toward the solution of groundwater problems [17].

Incorporating multiple data sources, such as satellite images, maps, tables, and other in situ data, provides the necessary reference information for analyzing accessible groundwater for scientists, researchers, and stakeholders [18–20]. One of the prevalent and reliable methods for identifying groundwater's accuracy is using a combination of remote sensors, GIS, and AHP [21]. The GIS tool is widely used in groundwater studies because it can visualize complex datasets [22]. AHP is a structured technique that analyzes multiple parameters to find solutions for the earth's environmental problems based on decision-making factors by assigning weights according to criteria [23–25]. The AHP technique is suitable for evaluating the consistency of the results, reducing possible errors in decision-making [23]. Many studies have successfully identified groundwater using remote sensors and multi-criteria AHP techniques [13,16].

In addition, the Caplina basin has an aquifer that presents a water table at an average depth of 50 m. This aquifer is lithologically made up of gravel, sand, silt, and clay. Furthermore, several studies on water quality monitoring were carried out in the Caplina basin [10,26,27]. These investigations sought to analyze the hydrodynamic, hydro-chemical, and hydrogeochemical characteristics to define the behavior and factors present in the Caplina aquifer, which is a vital support for water resource management for the inhabitants of the areas. However, these studies did not focus on delimiting the extent of groundwater at the basin level, nor did they apply remote sensing technology as an exploration and research method.

So, the objective of the study is to propose a model for determining the potential of groundwater based on the AHP-GIS methodology and validate the results obtained through information from existing wells, thus offering an accurate and reliable study that will help alleviate the problems of degradation of groundwater quality in the Caplina basin.

The results, in combination with remote sensing data, available maps, and the multi-criteria decision-making AHP method in conjunction with the overlay method in GIS, will help to answer the research questions such as the following: What would indicate the groundwater potential in arid zones? Can the AHP technique be combined with remote sensing analyses in a GIS environment to predict potential zones of groundwater and be useful for water resource management in arid regions? What parameters have more influence in determining groundwater potential?

## 2. Study Area

### 2.1. Location and Climate

The Caplina basin in southern Peru is located in the Atacama desert, characterized by a hyper-arid climate with little precipitation [4,28–32]; it has elevations ranging from 0 to 5660 m above sea level and an approximate total area of 4230 km$^2$. It is bordered to the west by the Sama basin, while on the east, it borders the Concordia basin, shared by Peruvian and Chilean territory, and the Lluta basin, which belongs to Chilean territory (Figure 1). The Caplina basin has two types of climate according to its location: a warm-temperate, desert-like climate with a moderate temperature range in the areas near the coast and a cold-moist climate in the high-elevation parts of the basin [33–35]. The temperature in the lower basin is 19.5 °C with scarce precipitation of around 6 mm/year. In contrast, in the high areas, the temperature is 4 °C with precipitation of around 350 mm/year [33].

### 2.2. Geological Setting

Rock outcroppings in the study area are controlled by erosion from the Western Cordillera and the Coastal Cordillera [36,37]. The rock outcroppings range in age from the Proterozoic to the Quaternary. However, the most abundant rocks are sedimentary and volcanic sedimentary from the Cenozoic, located from the center to the northeast of the basin. Five main formations have been identified: Sotillo (Paleocene), Lower Moquegua (Eocene),

Upper Moquegua (Oligocene), Huaylillas (Miocene), and Millo (Pliocene) [38,39]. The sedimentary rocks consist of conglomerates, reddish sandstones, and limestones originating from the erosion of the Coastal Basal Complex (Proterozoic), the Challaviento Intrusive Unit (Paleogene), the Ambo Group (Carboniferous), and the Chocolate Formation (Lower Jurassic). The volcanic sedimentary rocks consist of extensive deposits of ignimbrites that are linked to the activity of the Miocene volcanic arc (Huaylillas Volcanic Arc) [40]. The most recent sedimentary deposits belonging to the Quaternary are mostly found in the southwestern part of the basin. These sediments are formed by the latest periods of deposition in the softer parts of the basin and include river, alluvial, marine, eolian, eluvial, fluvio-glacial, morainic, and peat deposits. These Quaternary deposits are underground water storage, as in the Caplina aquifer [33].

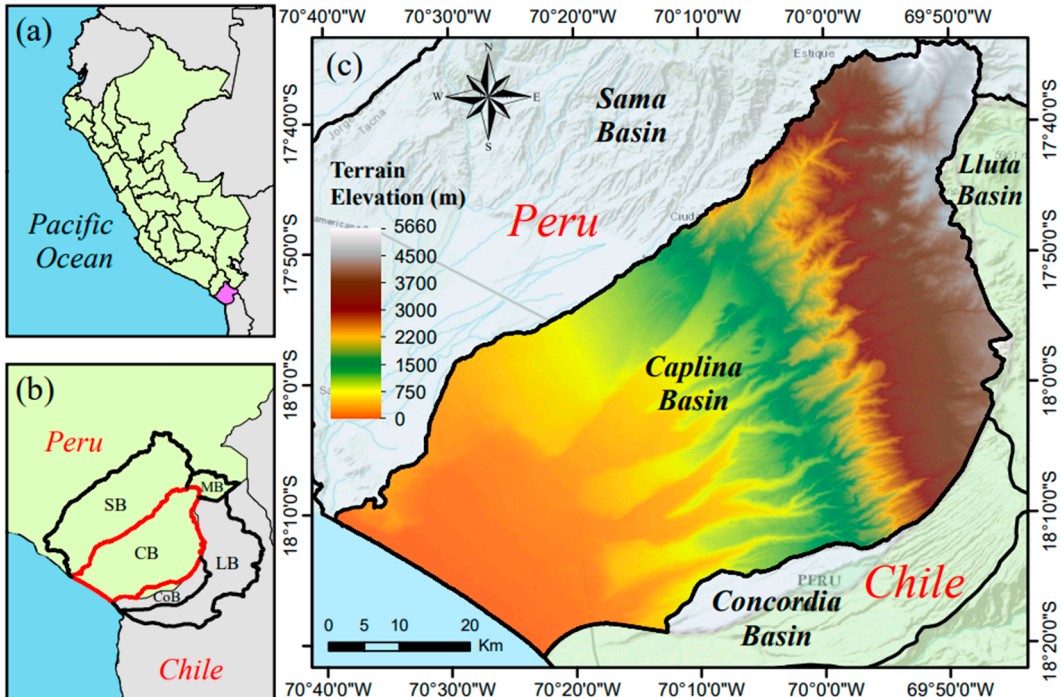

**Figure 1.** Location map of the study area (**a**) Tacna, Peru. (**b**) Adjacent basins, SB: Sama basin, MB: Maure basin, LB: Lluta basin, CoB: Concordia basin and CB: Caplina basin. (**c**) Geographical location of the Caplina basin, Atacama Desert.

## 3. Materials and Methods

### 3.1. Methodology

To determine the presence of groundwater in an area, several factors affecting its distribution and recharge should be evaluated. In this study, the variables related to geology, geomorphology, soil type, drainage density, lineament density, topographic moisture index, slope, ground cover, accumulated precipitation, and curvature were analyzed as factors that play an important role in defining the location of groundwater. Using the multi-criteria decision-making AHP method in conjunction with the overlay method in GIS, the final GWPZs map can be obtained. Finally, this map was validated using real data in the study area (Figure 2).

### 3.2. Datasets and Sources

Remote sensors continue to increase their technological development over the years; they are an ideal tool for various areas related to hydrology, geochemistry, and studies where field data are lacking or inadequate [41]. The delineation of groundwater is possible through the use of satellite data. In this study, obtaining thematic layers was performed using remote sensing technology, which is free to use and download. All downloaded

sources have been re-projected into the WGS84/UTM Zona 19S coordinate system. The necessary information was obtained from each remote sensor to generate the thematic layers (Table 1). The layers of drainage density, lineament density, slope, curvature, and topographic moisture index were generated from the 30 m resolution Digital Elevation Model (DEM) from the Shuttle Radar Topography Mission (SRTM). The land cover layer was obtained from the Sentinel-2 Land Use/Land Cover data with a resolution of 10 m. This is derived from Sentinel-2 images, which have been applied to classification models using artificial intelligence, allowing for a map of 9 surface classes to be obtained. The accumulated precipitation layer was obtained from the Terra-Climate satellite data, with a resolution of 4638 m; this sensor provides a set of monthly climate and climatic water balance data for global land surfaces [42].

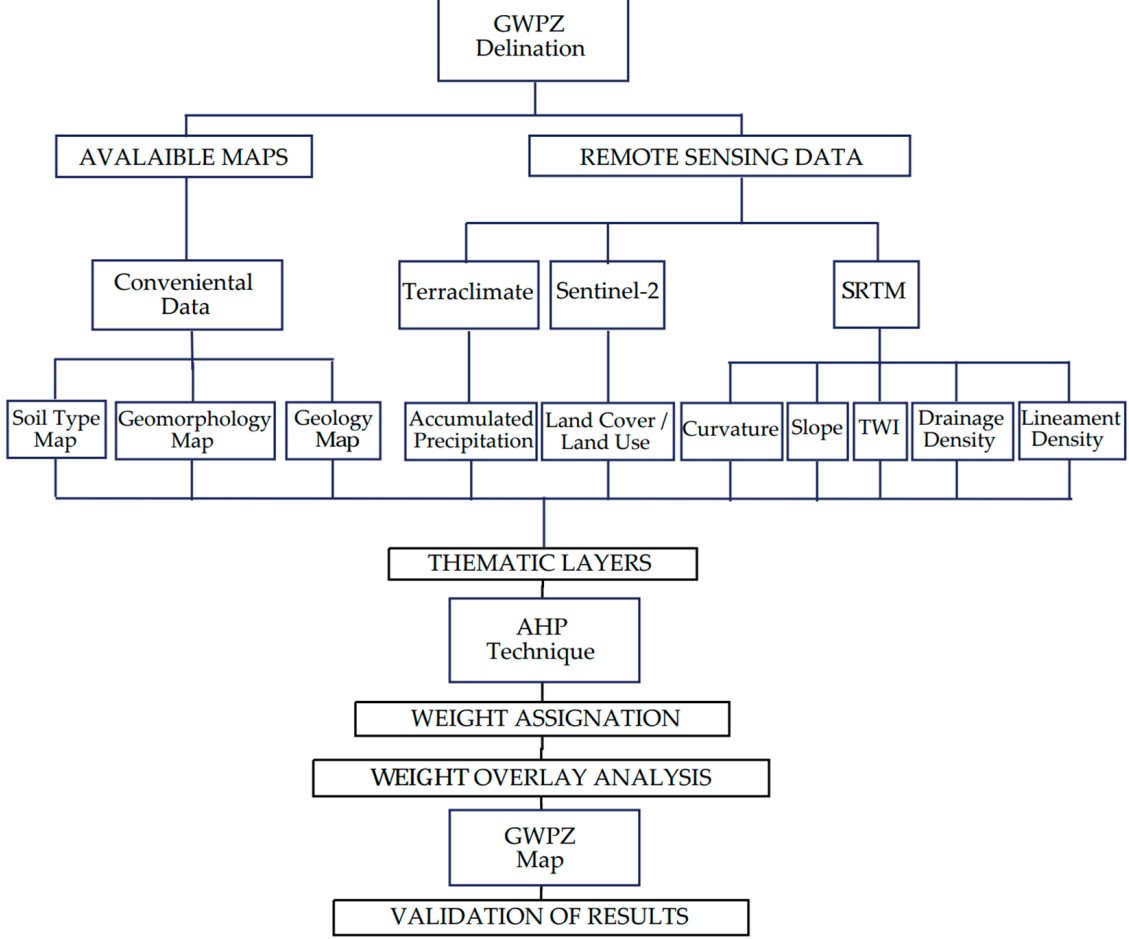

**Figure 2.** Flowchart of the methodology used for GWPZ mapping.

**Table 1.** Datasets used and their associated thematic layer.

| Dataset | Source | Thematic Layers |
|---|---|---|
| Shuttle Radar Topography Mission | USGS (https://earthexplorer.usgs.gov/, accessed on 15 January 2020) | Lineament density, Drainage density, Topographic wetness index, Slope, Curvature |
| Sentinel-2 LU/LC | ArcGIS (https://www.arcgis.com/home/ accessed on 20 January 2020) | Land use/Land cover |
| Terraclimate | Google Earth Engine (https://developers. google.com/earth-engine/datasets, accessed on 21 January 2020) | Annual accumulated precipitation |
| Geology | Geological, Mining and Metallurgical Institute | Lithological formations |
| Geomorphology | Geological, Mining and Metallurgical Institute | Geomorphology units |
| Soil | Ministry of the Environment | Soil type |

### 3.3. Weight Assignment Using AHP

The Analytic Hierarchy Process (AHP) is a Multi-Criteria Decision-Making (MCDM) analysis method proposed by Saaty in 1980 [23]; the method is based on the use of pairwise comparison matrices (PCM), which compare the general criteria with each other to estimate the range or weight of each criterion [24,25]. A hierarchy level is established among the ten proposed thematic layers to perform the pairwise comparison matrix. To determine the relative importance between the layers, the Saaty scale based on numerical values from 1–9 is used (Table 2), where 1 represents low importance among the evaluated layers, while 9 has highly favorable importance. In this study, to establish which layer has a more significant influence than the others, the impact that these have on the groundwater potential is analyzed, based on research results that are available in the area. Then to reduce the associated subjectivity caused by the inconsistency in establishing personal criteria, the resulting weights must be normalized from the pairwise comparison matrix (Table 3). Then, the following equation is used.

$$W_n = (\text{GM}/\sum_{n=1}^{N_f} GM_n) \tag{1}$$

where $GM_n$ represents the geometric mean of the n-th row of the matrix, which is calculated as follows.

$$\text{GM} = \sqrt[N_f]{A_{1n}A_{2n}\ldots A_{Nf}} \tag{2}$$

where $A_{Nf}$ represents the relative scale weight of the pairwise factors. Once the normalized weights (Table 4) have been obtained, the reliability of the result must be evaluated. To do this, the Consistency Ratio (CR) value is evaluated, which controls the balance between the assigned weights. The CR equation is as follows.

$$\text{CR} = \frac{\text{Consistency Index (CI)}}{\text{Random Index (RI)}} \tag{3}$$

**Table 2.** Saaty's scale of relative importance.

| Scale | Definition | Explanation |
|---|---|---|
| 1 | Equal importance | Two elements contribute equally to the objective. |
| 3 | Moderate importance | Experience and judgment slightly favor one element over another. |
| 5 | Strong importance | Experience and judgment strongly favor one element over another. |
| 7 | Very strong importance | One element is favored very strongly over another. |
| 9 | Extreme importance | The evidence favoring one element over another equals the highest possible order of affirmation. |
| 2,4,6,8 | Intermediate values | When compromise is needed. |

The matrix's Consistency Index (CI) is calculated using the following equation.

$$\text{CI} = \frac{\lambda_{max} - n}{n - 1} \tag{4}$$

where $\lambda_{max}$ is the maximum eigenvalue of the matrix and $n$ is the number of criteria; thus, if the condition CR < 1.0 is met, the model is acceptable. On the other hand, it is necessary to re-evaluate the criteria used when assigning weights. The study shows a CR value of 0.065, which is considered an acceptable consistency.

### 3.4. Weighted Overlay Integration

The thematic layers have been joined using the weighted overlay analysis method applied in the GIS environment to generate the groundwater potential map. To calculate the probability index of groundwater occurrence, the following equation is used:

$$\text{GWPZ} = \sum_{i=1}^{n} \sum_{j=1}^{m} W_i * X_j \tag{5}$$

where $W_i$ expresses the normalized weight of the i-th thematic variable, $X_j$ expresses the normalized weight of the j-th class of the variable, n represents the total number of variables, and m represents the total number of classes of a variable. The results obtained for GWPZ were grouped into four classes: high, moderate, low, and very low.

**Table 3.** Pairwise comparison matrix of the thematic layers.

|     | GL | SL | GM | AP | LU | SO | LD | DD | CV | TWI |
|-----|------|------|------|-------|-------|-------|-------|-------|-------|-------|
| GL | 1 | 2.00 | 3.00 | 3.00 | 5.00 | 9.00 | 6.00 | 7.00 | 8.00 | 9.00 |
| SL | 0.50 | 1 | 2.00 | 4.00 | 5.00 | 7.00 | 5.00 | 7.00 | 6.00 | 8.00 |
| GM | 0.33 | 0.50 | 1 | 3.00 | 4.00 | 5.00 | 7.00 | 6.00 | 7.00 | 8.00 |
| AP | 0.33 | 0.25 | 0.33 | 1 | 2.00 | 3.00 | 5.00 | 7.00 | 6.00 | 8.00 |
| LU | 0.20 | 0.20 | 0.25 | 0.50 | 1 | 2.00 | 3.00 | 4.00 | 3.00 | 4.00 |
| SO | 0.11 | 0.14 | 0.20 | 0.33 | 0.50 | 1 | 2.00 | 3.00 | 3.00 | 4.00 |
| LD | 0.17 | 0.20 | 0.14 | 0.20 | 0.33 | 0.50 | 1 | 2.00 | 2.00 | 4.00 |
| DD | 0.14 | 0.14 | 0.17 | 0.14 | 0.25 | 0.33 | 0.50 | 1 | 3.00 | 4.00 |
| CV | 0.12 | 0.17 | 0.14 | 0.17 | 0.33 | 0.33 | 0.50 | 0.33 | 1 | 2.00 |
| TWI | 0.11 | 0.12 | 0.12 | 0.12 | 0.25 | 0.25 | 0.25 | 0.25 | 0.50 | 1 |
| Sum | 3.02 | 4.73 | 7.36 | 12.47 | 18.67 | 28.42 | 30.25 | 37.58 | 39.50 | 52.00 |

Notes: Abbreviations: GL = geology; SL = slope; GM = geomorphology; AP = accumulated precipitation; LU = land use; SO = soil; LD = lineament density; DD = drainage density; CV = curvature; TWI = topographic wetness index.

**Table 4.** Normalized weight of the matrix.

|     | GL | SL | GM | AP | LU | SO | LD | DD | CV | TWI | $N_{wt}$ % |
|-----|------|------|------|------|------|------|------|------|------|------|------|
| GL | 0.331 | 0.423 | 0.408 | 0.241 | 0.268 | 0.317 | 0.198 | 0.186 | 0.203 | 0.173 | 28.00 |
| SL | 0.165 | 0.212 | 0.272 | 0.321 | 0.268 | 0.246 | 0.165 | 0.186 | 0.152 | 0.154 | 22.40 |
| GM | 0.110 | 0.106 | 0.136 | 0.241 | 0.214 | 0.176 | 0.231 | 0.160 | 0.177 | 0.154 | 17.40 |
| AP | 0.110 | 0.053 | 0.045 | 0.080 | 0.107 | 0.106 | 0.165 | 0.186 | 0.152 | 0.154 | 11.20 |
| LU | 0.066 | 0.042 | 0.034 | 0.040 | 0.054 | 0.070 | 0.099 | 0.106 | 0.076 | 0.077 | 6.50 |
| SO | 0.037 | 0.030 | 0.027 | 0.027 | 0.027 | 0.035 | 0.066 | 0.080 | 0.076 | 0.077 | 4.60 |
| LD | 0.055 | 0.042 | 0.019 | 0.016 | 0.018 | 0.018 | 0.033 | 0.053 | 0.051 | 0.077 | 3.50 |
| DD | 0.047 | 0.030 | 0.023 | 0.011 | 0.013 | 0.012 | 0.017 | 0.027 | 0.076 | 0.077 | 2.90 |
| CV | 0.041 | 0.035 | 0.019 | 0.013 | 0.018 | 0.012 | 0.017 | 0.009 | 0.025 | 0.038 | 2.10 |
| TWI | 0.037 | 0.026 | 0.017 | 0.010 | 0.013 | 0.009 | 0.008 | 0.007 | 0.013 | 0.019 | 1.50 |
| Sum | 1.000 | 1.000 | 1.000 | 1.000 | 1.000 | 1.000 | 1.000 | 1.000 | 1.000 | 1.000 | 100.00 |

Note: Abbreviations: $N_{wt}$ % = normalized weight.

## 4. Results

### 4.1. Thematic Layer Description

#### 4.1.1. Geology

Superficial geological structural and hydraulic characteristics are crucial in controlling the interaction between the surface and groundwater in arid climates [43–47]. Similarly, lithological properties determine groundwater porosity and movement [48]; if the lithology's porosity is higher, it tends to produce more groundwater storage. The geological data have been obtained from the Geological, Mining, and Metallurgical Institute (INGEMMET).

The basin geology encompasses formations from the Proterozoic to Quaternary sedimentary deposits (Figure 3a). The present units that cover the largest areas are the Huaylillas formation and alluvial deposits. The Huaylillas formation, which consists of rhyolitic and dacitic tuffs, covers 34.7% of the total area.

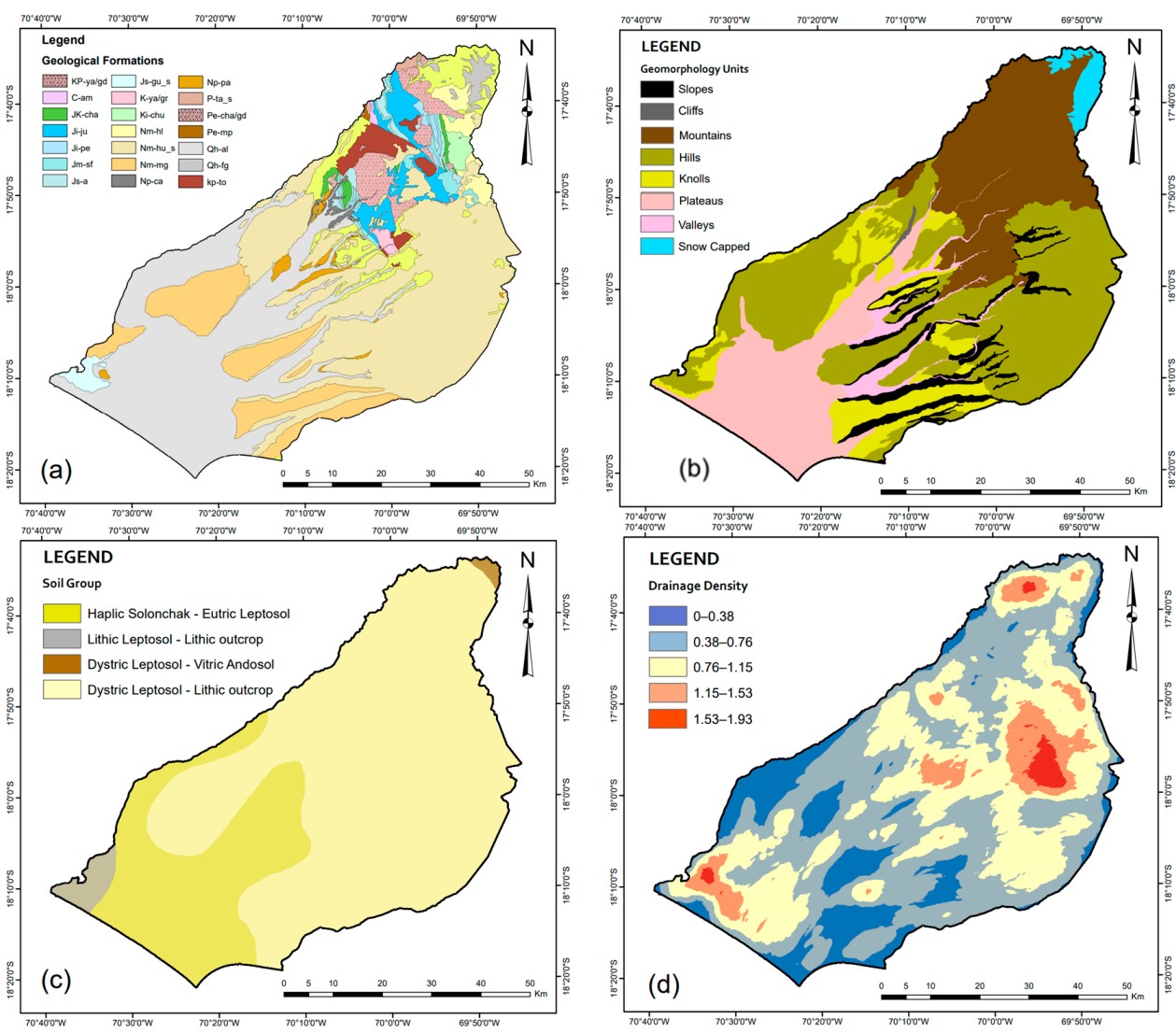

**Figure 3.** (**a**) Geology, (**b**) geomorphology, (**c**) soil, and (**d**) drainage density map.

Meanwhile, alluvial deposits cover 30% of the total area, covering much of the low-slope areas of the basin. Due to their high permeability and porosity, Sedimentary rocks and alluvial deposits favor the recharge rate of groundwater and, therefore, were assigned the highest weight. On the other hand, volcanic and intrusive rocks were assigned the lowest weight (Table 5).

### 4.1.2. Geomorphology

The geomorphological features of a basin play a crucial role in delimiting the groundwater potential. The area's geomorphology has a more tangible character and a direct connection between the movement of groundwater in the subsurface [49], which characterizes the potential of groundwater according to its given form [50]. The geomorphology information was obtained from the map published by the Geological, Mining, and Metallurgical Institute (INGEMMET) [47]. Geomorphologically, the basin presents hills, cliffs, knolls, mountains, snow-capped peaks, plateaus, slopes, and valleys (Figure 3b). The most abundant geomorphological units are hills, plateaus, and mountains. Hills cover 1658 km²

of the total area, predominating in the southwest and middle of the basin. Mountains occupy 946 km$^2$ of the total area, located on the northeast side of the basin. Finally, the plateaus cover 713 km$^2$ of area, mostly near the coast and middle of the basin. Greater weight was assigned to the plateaus, while lower weight was assigned to the areas of high slope such as mountains and slopes (Table 5).

**Table 5.** Weight assignment and ranking of the subclasses for the thematic layers.

| Thematic Layer | Assigned Weight | Classes | Rank |
| --- | --- | --- | --- |
| Geology | 0.28 | Sedimentary rocks | 4 |
| | | Volcanic rocks | 1 |
| | | Volcanic sedimentary | 2 |
| | | Intrusive rocks | 1 |
| | | Alluvial deposits | 5 |
| | | Fluvio-glacial deposits | 3 |
| Geomorphology | 0.17 | Hills | 2 |
| | | Escarps | 2 |
| | | Mountains | 1 |
| | | Snow covered | 4 |
| | | Plains | 5 |
| | | Slope | 1 |
| | | Valleys | 4 |
| | | Knolls | 3 |
| Soil | 0.04 | SCh–LPe | 1 |
| | | LPq–R | 3 |
| | | LPd–ANz | 4 |
| | | LPd–R | 5 |
| Drainage density | 0.02 | 0–0.38 | 5 |
| | | 0.38–0.76 | 4 |
| | | 0.76–1.15 | 3 |
| | | 1.15–1.53 | 2 |
| | | 1.53–1.93 | 1 |
| Lineament density | 0.03 | 0–0.77 | 1 |
| | | 0.77–1.55 | 2 |
| | | 1.55–2.32 | 3 |
| | | 2.32–3.10 | 4 |
| | | 3.10–3.88 | 5 |
| TWI | 0.01 | −3.53–2.35 | 1 |
| | | 2.35–8.23 | 2 |
| | | 8.23–14.11 | 3 |
| | | 14.11–19.98 | 4 |
| | | 19.98–25.89 | 5 |
| Slope | 0.22 | 0–3 | 5 |
| | | 3–7 | 4 |
| | | 7–14 | 3 |
| | | 14–30 | 2 |
| | | 30–75 | 1 |
| Curvature | 0.03 | −0.046−−0.009 | 1 |
| | | −0.009−−0.002 | 2 |
| | | −0.002–0.002 | 3 |
| | | 0.002–0.008 | 4 |
| | | 0.008–0.022 | 5 |
| Precipitation | 0.11 | 0–40.37 | 1 |
| | | 40.37–119.29 | 2 |
| | | 119.29–209.22 | 3 |
| | | 209.22–313.83 | 4 |
| | | 313.83–468 | 5 |

**Table 5.** *Cont.*

| Thematic Layer | Assigned Weight | Classes | Rank |
|---|---|---|---|
| LU/LC | 0.06 | Waterbodies | 4 |
| | | Trees | 2 |
| | | Grass | 4 |
| | | Cropland | 5 |
| | | Shrubland | 3 |
| | | Urban Area | 2 |
| | | Bare Land | 1 |
| | | Snow | 4 |

### 4.1.3. Soil

The soil types play an important role in the amount of water that can infiltrate the subsurface formations and thus affect groundwater recharge [51,52]. The infiltration rate largely depends on the soil texture and hydraulic characteristics of the soil. Fine-grained soils have limited infiltration due to relatively low permeability, unlike coarse-grained soils, where water easily infiltrates due to their high permeability [13]. The soil data were obtained from the Ministry of the Environment (MINAM). The Caplina basin presents four soil types (Figure 3c), District Leptosol–Lithic Outcropping (LPd-R), District Leptosol–Vitric Andosol (LPd-ANz), Lithic Leptosol–Lithic Outcropping (LPq-R), and Haplic Solonchak–Eutric Leptosol (SCh-LPe). LPd-R covers the largest area of the basin, with 72% of the total area equalling 3036 km$^2$. It is followed by the SCh-LPe soil, which covers 25.5% of the entire area. Due to its higher infiltration capacity relationship, higher weight was assigned to the SCh-LPe and lower weight to the LPd-ANz (Table 5).

### 4.1.4. Drainage Density

The drainage density expresses the proximity of spacing in the channels of streams and, therefore, provides a quantitative measure of the average length of these [53]. Drainage density is important in evaluating groundwater potential and has an inverse relationship with permeability [54]. To obtain the results, work was performed with the DEM in the QGIS environment using the Line Density interpolation tool. The processing results show density ranges of 0–1.93 km$^2$. The values were then reclassified, grouping them into five new subclasses (Figure 3d): very low (0–0.38 km$^2$), low (0.38–0.76 km$^2$), medium (0.76–1.15 km$^2$), high (1.15–1.53 km$^2$), and very high (1.53–1.93 km$^2$). The areas of very low drainage density are related to a high infiltration ratio, thus giving a positive potential for groundwater. Therefore, it has the highest weight. Conversely, the exceptionally high drainage density area was assigned the lowest weight (Table 5).

### 4.1.5. Lineament Density

The lineament is considered an essential geomorphological feature as a tool for studying the prediction of groundwater resources. The presence of these lineaments in hard rock terrain is indicative of the presence of good groundwater storage [55]. These lineaments serve as conduits for the movement and storage of groundwater in the aquifer [56]. The lineament map was obtained using the DEM and the Lineament Density tool in the QGIS environment, and the lineament density ranges from 0 to 3.88 km$^2$. These values were reclassified into five subclasses (Figure 4a): very low (0–0.77 km$^2$), low (0.77–1.55 km$^2$), medium (1.55–2.32 km$^2$), high (2.32–3.10 km$^2$), and very high (3.10–3.88 km$^2$). The very high lineament densities are located in the middle part of the basin, and the highest weight was assigned to this classification. In contrast, the very low densities were assigned the lowest weight (Table 5).

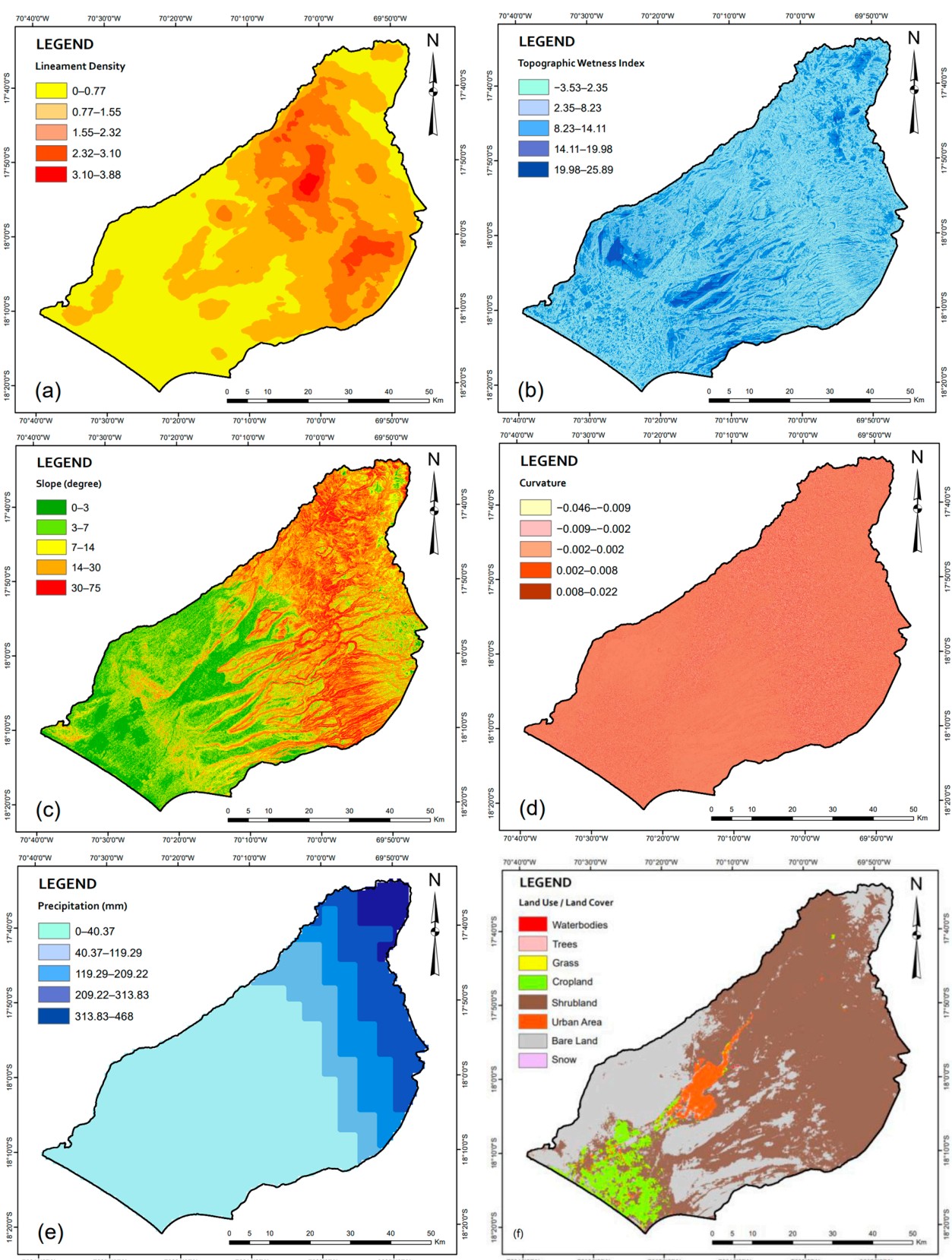

**Figure 4.** (**a**) Lineament density, (**b**) topographic wetness index, (**c**) slope, (**d**) curvature, (**e**) precipitation, and (**f**) LU/LC map.

### 4.1.6. Topographic Wetness Index

The Topographic Wetness Index (TWI) is commonly used to calculate topographic control of hydrologic processes and reflects the potential for underground water infiltration caused by topographic effects [57]. Consequently, areas with a high TWI have a high potential for underground water, while areas with a low TWI have a low potential for underground water [58]. The TWI map was obtained in the QGIS environment using the Raster Calculator tool by applying the following equation:

$$\text{TWI} = \ln\left(\frac{\alpha}{\tan\beta}\right) \tag{6}$$

where $\alpha$ represents the specific catchment area and $\beta$ represents the slope value. The result of TWI in the study area ranges from −3.53 to 25.89. The TWI layer has been reclassified into five subclasses (Figure 4b): very low (−3.53–2.35), low (2.35–8.23), medium (8.23–14.11), high (14.11–19.98), and high (19.98–25.89). The very high class was assigned the highest weight, while the very low class was assigned the lowest weight (Table 5).

### 4.1.7. Slope

The slope has a positive relationship with runoff and, therefore, has an inverse influence on the infiltration and recharge capacity. With the increase of slope angle values, the high velocity of the surface flow will increase [59,60]. Higher slopes produce less recharge because the water received from precipitation quickly flows down a steep slope during rain. Thus, it needs more residence time to infiltrate and recharge the saturated zone [61]. The slope map is the result of applying the Slope terrain analysis tool on the DEM in the QGIS environment. The study area presents slopes between 0° and 75° (Figure 4c). A reclassification of values was performed, grouping them into five subclasses, flat slope areas (0°–3°), gentle (3°–7°), medium (7°–14°), strong (14°–30°), and very strong (30°–75°). The flat and gentle slope areas have a higher infiltration ratio. Thus, they were assigned the highest weight, and the very strong slope areas were assigned the lowest weight (Table 5).

### 4.1.8. Curvature

Curvature measures the intensity of surface convexity and concavity reflected in positive and negative values, respectively. Curvature can provide an estimate of infiltration conditions [62]. A concave slope can retain more water for a longer period of rain [63–65]. To obtain the slope layer, the Curvature terrain analysis tool was applied to the DEM in the QGIS environment, resulting in curvature ranges from −4.6 to 2.1 (Figure 4d). The values were reclassified into five subclasses: very low curvature (−4.6–−0.9), low (−0.9–−0.2), medium (−0.2–0.2), high (0.2–0.8), and very high (0.8–2.1). The highest weight corresponds to the highest curvatures, while the lowest weight corresponds to the lowest curvatures (Table 5).

### 4.1.9. Accumulated Precipitation

Rain is one of the main components of the hydrological cycle and is the main source of recharge for groundwater [66]. There is a strong positive correlation between precipitation and groundwater levels, as the fluctuation of the water table is influenced by precipitation in the basin. A region with high rainfall levels is considered an area with higher water potential and vice versa [67]. The accumulated precipitation layer was obtained from the climate and annual water balance data provided by the Terraclimate remote sensor. In the study area, the obtained data show the presence of precipitation between 0 mm and 468 mm/year (Figure 4e). High precipitation levels are located in the northwest of the basin, associated with high elevation areas; therefore, the highest weight was assigned to it. On the other hand, low precipitation levels are in low-elevation areas; thus, the lowest weight was assigned to them (Table 5).

#### 4.1.10. LU/LC

Land use affects groundwater through changes in recharge and demand. Furthermore, groundwater is closely related to landscape and land use [68]. The volume, time, and amount of groundwater recharge are controlled by land cover, affecting runoff and evapotranspiration [69]. The LU/LC layer was obtained from Sentinel-2 10 m Land Use/Land Cover images. The study area has nine land cover classes: water, urban areas, trees, bogs, grass, crop area, shrubs, snow, and bare soil (Figure 4f). Shrubs are the most extensive unit in the basin, covering 62% of the total area, followed by bare soil with 29.7%. The highest weight values were assigned to the crop areas and water bodies due to their notable association with groundwater recharge. Bare soil areas negatively affect potential groundwater areas (Table 5).

#### 4.2. Groundwater Potential Zone Map

The concept of AHP as a multi-criteria decision-making method was introduced by Saaty in the 1980s [23]. Several decades later, this methodology has evolved to reach its application in detailed study branches such as hydrology, geochemistry, and earth science-related disciplines [63–65]. The qualities of AHP as a working method, in conjunction with a QGIS environment, show positive results in validation with the statistics and data used. To establish the possible potential groundwater zones, the influence of 10 thematic layers was analyzed: geology, geomorphology, soil type, drainage density, lineament density, topographic moisture index, slope, soil coverage, accumulated precipitation, and curvature. The reliability of the criteria used has been evaluated by analyzing the Consistency Ratio, showing an acceptable value of CR = 0.065 (Figure 5). Similarly, for validation, the Receiver Operating Characteristic (ROC) curve was used, showing positive results that reflect the real data of the study area.

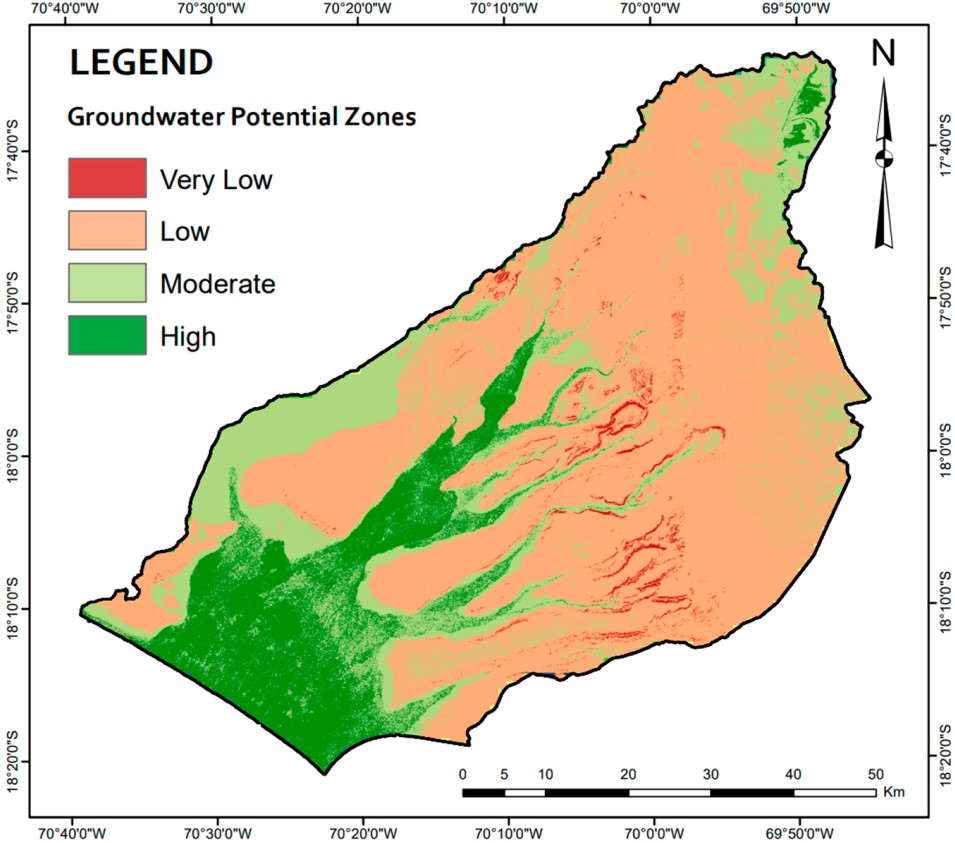

**Figure 5.** Groundwater potential zones map.

Identification of groundwater has become a global necessity in recent decades [44]. However, the Caplina basin, due to its arid climate, faces a constant water shortage issue and requires sustainability projects for its potential water resources. The GWPZ map shows four classifications: high, moderate, low, and very low potential (Figure 5). High potential zones are associated with low slope characteristics, permeable lithologies, and low drainage densities. The high potential zones are mostly located in the southwest and small areas northeast of the Caplina basin, covering 696 km$^2$ (15.02%) of the total area [70,71]. In the case of moderate potential zones, they cover 1111 km$^2$ (23.93%) of the basin; they are dispersed throughout the basin representing a lower relevance of groundwater. The low and very low potential zones occupy 2776 km$^2$ (59.80%) and 58 km$^2$ (1.25%), respectively. They are the most abundant groups in the basin. These zones are characterized by high elevations, non-porous lithologies, and low groundwater recharge rates (Table 5). The study then shows a clear relationship between geology, geomorphology, and slope as the main decisive factors in the location of groundwater. Furthermore, other studies in this study area concur with the mentioned factors as indicators of the occurrence and recharge of groundwater [14,28,40,66].

*4.3. Validation*

The validation stage is a fundamental requirement to establish the accuracy of a generated model. The GWPZ map has been validated through correlation analysis using the data of existing water wells and natural springs in the Caplina basin. The well and spring data used was obtained from the National Water Authority (ANA) database, which includes a total of 170 records that have been located on the resulting GWPZ map (Figure 6a). The ROC curve for the Caplina basin model (Figure 6b) shows a value of Area Under the Curve (AUC) of 0.869, representing 86.9% accuracy. According to the obtained AUC value, the prediction of the GWPZ map is considered satisfactory. In this way, the model presents a moderate to the high predictability of groundwater thanks to the manipulation of remote sensor data in conjunction with the AHP technique.

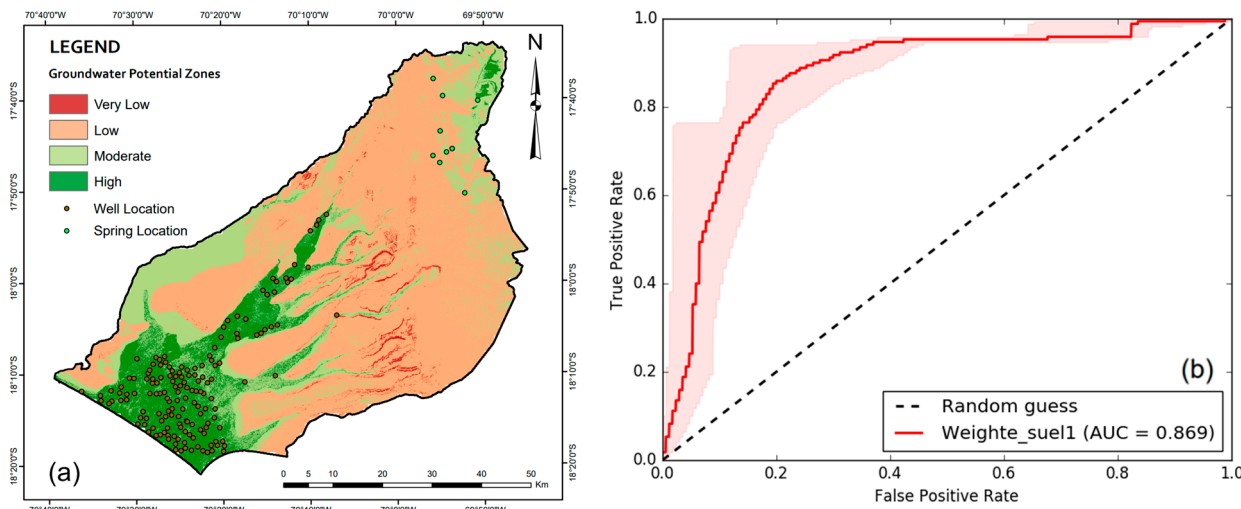

**Figure 6.** (**a**) Well and spring locations used for the model validation. (**b**) Receiver Operating Characteristic (ROC) curve for groundwater potential map validation.

The GWPZ map was also validated through comparison with the existing exploitation volume map developed by the Ministry of Agriculture and Risk in 2016 (Figure 7). The exploitation volume map expresses the amount of water extracted from wells; these measurements range from the minimum of 44 m$^3$/year to the maximum of 2,515,314 m$^3$/year of exploitable water. Upon comparison, it is observed that the central zone of the aquifer presents the highest values of exploitation volumes per year and, in turn, presents the highest concentration of high groundwater potential. This positive relationship between

high potential and groundwater volume at the basin level is because they are fed by the Caplina river, which has an extensive course coming from the sectors of maximum precipitation. At the same time, the areas to the east of the aquifer show lower exploitation volumes per well and a slight predominance of high groundwater potential. This moderate relationship between potential and water volume is due to the weaker feeding from the rain from the Cauñani, Espiritus, and Escritos (Figure 7). From the validation result, it is feasible to observe that the GWPZ model presents a logical relationship with the distribution of exploitation volumes. In this way, the different validation classes applied to the GWPZ model demonstrate an appropriate association of the delineated zones with the real data of the study area.

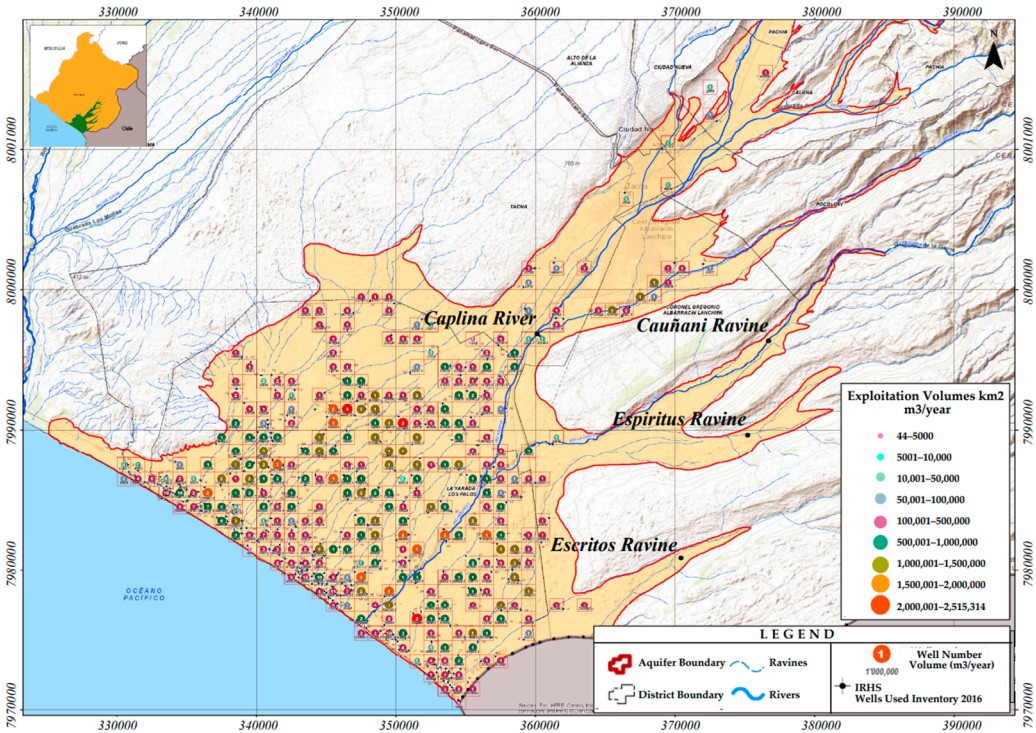

**Figure 7.** Exploitation volumes map of the Caplina aquifer, 2016. Modified from the Ministry of Agriculture and Irrigation (MIDAGRI).

## 5. Discussion

Several surrounding factors have been combined to obtain the groundwater potential map. The parameters of geology, geomorphology, and slope have the greatest influence in determining groundwater potential. On the other hand, TWI, drainage density, and lineament density present the lowest weights, reflecting a relationship with the low rainfall presence in the study area.

The GWPZ map shows that the areas with high potential are mostly in the southwestern part of the basin, near the coast. This indication has a favorable relationship with the validation of the results since near these low-elevation areas there is the highest number of exploited wells throughout the basin (Figure 6a). The set of factors of flat slopes, geomorphological units such as plains, and alluvial gravel and sand deposits, allow for the highest occurrence of groundwater potential. Similarly, urban areas such as Tacna have a lower amount of high potential, where the exploitation of wells is more limited (Figure 6a). Similarly, a small area in the northeast of the basin has also been identified as having a high groundwater potential, far from the previously mentioned two zones. This area is in the western cordillera, where the greatest precipitation occurs with favorable geology and moderate slopes, forming an appropriate zone of high groundwater potential.

There is currently much controversy in quantifying the recharge of the Caplina aquifer; this leads to studies to identify areas of high groundwater potential. Thus, this proposed model will be very useful to update and calibrate existing hydrogeological models in the Caplina aquifer. The delimitation of areas with poor and very poor groundwater potential is much more extensive than the favorable areas [72]. They mainly dominate the center and northeast of the basin; they are associated with barren soil with non-permeable lithologies. These are unfavorable areas on the eastern side of the basin due to the extensive surface covering of the Huaylillas volcanic sedimentary formation along with high elevations and steep slopes [39]. These areas are not feasible for exploitation and, together with low precipitation rates, do not allow for high groundwater recharge.

According to the validation of the results, the AHP technique, combined with the analysis of remote sensors in a GIS environment, allow for the precise and adequate prediction of potential zones of groundwater. An advantage to highlight of the applied methodology is its flexibility to adapt to other areas of study [21]. The change of variables and their possible influence is optional, depending on the characteristics of the new study area. The results obtained from this study are open to discussion in future research. The active evolution of remote sensors means that new data are increasingly detailed, allowing for the progressive improvement of the accuracy of the groundwater potential map.

In recent years, new alternative methodologies have been developed from the classic concept applied in the AHP. Among these is Fuzzy AHP, which plays the role of modeling and simulation technique for complex systems, such as spatial objects on a map [73,74]. The TOPSIS method gives an approach for the geometric calculation between the distances of each alternative with the ideal alternative. There are also other methods that implement different scientific concepts in decision-making. While these methods have grown in reputation over the years, the AHP method is still widely accepted by the community, due to its solid theoretical foundations and manageable simplicity. In the same way, the objective of this study had an approach aimed at evaluating the areas of high groundwater potential, opting for the use of a simple, reliable, and flexible method in the execution of the research.

## 6. Conclusions

The creation of the GWPZ map was made possible by incorporating remote sensors and open-access maps in a QGIS environment. This, together with the AHP method, yields accurate and reliable results, thus demonstrating that it is an appropriate tool for delimiting potential groundwater areas. This study shows that the Atacama Desert, despite its inherently arid climate, consists of four groups of potential zones (GWPZ): high (15.02%), moderate (23.93%), low (59.80%), and very low (1.25%). The parameters of geology, geomorphology, and slope have the greatest influence in determining groundwater potential in the Atacama Desert. Specifically, the results indicate that alluvial deposits on smooth slopes show a high groundwater potential. The methodology applied in this research can be adapted to different basins located in the Atacama Desert and other desert areas, thus seeking to increase its validation and applicability. Similarly, the result of the GWPZ map seeks to contribute as a management method for exploration based on well drilling projects and/or water resource management projects in the Caplina basin, playing a predominant role in the face of the latent problem of water scarcity in the study area. Finally, these results seek to expand the knowledge of the recharge behavior of the Caplina aquifer with the new areas marked by the model obtained.

**Author Contributions:** Conceptualization, E.P.-V., V.P., E.I.-B. and S.C.; software, V.P.; data curation, S.C.; validation, E.P.-V. and G.H.; formal analysis, V.P., G.H., S.C., E.I.-B. and E.P.-V.; writing—original draft preparation, V.P. and S.C.; writing—review and editing, V.P., G.H., S.C., E.I.-B. and E.P.-V.; project administration, E.P.-V. All authors have read and agreed to the published version of the manuscript.

**Funding:** This research received no external funding.

**Acknowledgments:** Funds financed this work from the mining royalties, IGIN, and VIIN of the UNJBG, within the research project "Study of Hydraulic Recharge and Salinization Processes in the Caplina Aquifer, Tacna, Peru, for a Sustainable Management of Groundwater".

**Conflicts of Interest:** The authors declare no conflict of interest.

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
