# Peer review of "A Multi-Criteria Decision-Making Technique Using Remote Sensors to Evaluate the Potential of Groundwater in the Arid Zone Basin of the Atacama Desert"

_water, doi:10.3390/w15071344_

Round 1

Reviewer 1 Report

This research work is meaningful and useful. The whole paper has strong logic, clear hierarchy, multiple and complete analysis factors, reliable data, and well verified results and conclusions, and good writing. So I recommend publishing after minor revision.

The places requiring explanation or minor rerision are as follows:

1. The authors are asked to explain the basis for the grading of each indicator?

2. How to define the grade comparison weight of each indicator?

3. The groundwater depth information in the study area need to be supplemented;

4. The letters "b" and "y" are in Lines 83 and 276, respectively. What do they mean or are they wrong?

5. The color of the legend in Figure 4f is not completely consistent with the figure, such as shrubland;

6. The references in Lines 368-369 are incorrect.

Author Response

Agradecemos la revisión del revisor, ya que nos permitió mejorar el trabajo

  1. Se pide a los autores que expliquen la base para la calificación de cada indicador.
  2. ¿Cómo definir el peso de comparación de calificaciones de cada indicador?

Respuesta:

En la línea 170, se incorporó un párrafo en el texto como se muestra a continuación:

A hierarchy level is established among the ten proposed thematic layers to perform the pairwise comparison matrix. To determine the relative importance between the layers, the Saaty scale based on numerical values from 1-9 is used (Table 2), where 1 represents low importance among the evaluated layers, while 9 has highly favorable importance. In this study, to establish which layer has a more significant influence than the others, the impact that these have on the groundwater potential is analyzed, based on research results that are available in the area. Then to reduce the associated subjectivity caused by the inconsistency in establishing personal criteria, the resulting weights must be normalized from the pairwise comparison matrix.

  1. The groundwater depth information in the study area need to be supplemented

Answer:

En la línea 70, agregamos la siguiente declaración:

Además, la cuenca Caplina cuenta con un acuífero que presenta un nivel freático a una profundidad promedio de 50 metros. Este acuífero está compuesto litológicamente por grava, arena, limo y arcilla.

  1. Las letras "b" e "y" están en las líneas 83 y 276, respectivamente. ¿Qué significan o están equivocados?

Respuesta:

La corrección gramatical se abordó en las líneas 83 y 276.

  1. El color de la leyenda en la Figura 4f no es completamente consistente con la figura, como matorrales

Respuesta:

La corrección de la leyenda se abordó en la Línea 320. :

  1. Las referencias en las líneas 368-369 son incorrectas

Respuesta:

Se corrigieron las referencias en la Línea 379

Reviewer 2 Report

The manuscript is of high quality and represents a very good study with an interesting methodology. The authors are very familiar with the fields. Contributions to the work are:

- The authors have applied existing studies and developed a certain methodology for improving individual problems.

- The paper has an excellent structure with.

- Steps in operation are solidly explained.

- The developed methodology is given in detail.

- Good problem definition.

The paper has great potential and can be accepted after the following MINOR corrections:

1. It is necessary to technically edit the text (especially in the case of setting the size of the tables to the text format; in Line 368, fix the indicated error about the inability to connect to the source?

2. In the conclusion, it is necessary to answer the questions posed in the introductory part i give clearer tasks for subsequent research with a connection to the obtained results.

3. I believe that the authors should emphasize more clearly why they chose AHP for research, and not other improved or newer methods. AHP is a technique developed in the 1970s, possibly reaching its goal but still quite old.
Other possible solutions could have been, for example:
- Fuzzy AHP which is able to classify evaluation factors into target level, criterion level, and factor level (while AHP can only target level and factor level)
- TOPSIS based on distance, with perhaps a clearer and mathematically sound structure,
- or hybrid MCDM approaches, like a combination AHP/TOPSIS when TOPSIS employs AHP weights, or Cybernetic, Parsimonious, Express Fuzzy AHP/ANP etc.
?

Author Response

We appreciate the review of the reviewer, as it allowed us to improve the paper

  1. It is necessary to technically edit the text (especially in the case of setting the size of the tables to the text format; in Line 368, fix the indicated error about the inability to connect to the source?

Answer:

It was solved in the text. The reference was corrected in Line 379

  1. In the conclusion, it is necessary to answer the questions posed in the introductory part give clearer tasks for subsequent research with a connection to the obtained results

Answer:

The conclusion of this study highlights that the AHP technique with remote sensing analysis in a GIS environment can predict potential groundwater in arid regions. In addition, it emphasizes the potential of groundwater in the arid areas of the Caplina aquifer. On the other hand, the parameters that have the greatest influence in determining the potential of groundwater are mentioned.

The conclusion is as follows:

The creation of the GWPZ map was made possible by incorporating remote sensors and open-access maps in a QGIS environment. This, together with the AHP method, yields accurate and reliable results, thus demonstrating that it is an appropriate tool for delimiting potential groundwater areas. This study shows that the Atacama Desert, despite its inherent arid climate, has four groups of potential zones (GWPZ), high (15.02%), moderate (23.93%), low (59.80%), and very low (1.25%). On the other hand, the parameters of geology, geomorphology, and slope have the greatest influence in determining groundwater potential in the Atacama Desert. In addition, The result indicates that alluvial deposits on smooth slopes show a high groundwater potential. The methodology applied in this research can be adapted to different basins located in the Atacama Desert and other desert areas, thus seeking to increase its validation and applicability. Similarly, the result of the GWPZ map seeks to contribute as a management method for exploration based on well drilling projects and/or water resource management projects in the Caplina basin, playing a predominant role in the face of the latent problem of water scarcity in the study area. Finally, these results seek to expand the knowledge of the recharge behavior of the Caplina aquifer with the new areas marked by the model obtained.  

  1. I believe that the authors should emphasize more clearly why they chose AHP for research, and not other improved or newer methods. AHP is a technique developed in the 1970s, possibly reaching its goal but still quite old.

Other possible solutions could have been, for example:

- Fuzzy AHP which is able to classify evaluation factors into target level, criterion level, and factor level (while AHP can only target level and factor level)

- TOPSIS based on distance, with perhaps a clearer and mathematically sound structure,

- or hybrid MCDM approaches, like a combination AHP/TOPSIS when TOPSIS employs AHP weights, or Cybernetic, Parsimonious, Express Fuzzy AHP/ANP etc.?

Answer:

The following paragraph was added at the end of the discussion. Line 468

In recent years, new alternative methodologies have been developed from the classic concept applied in the AHP. Among these is Fuzzy AHP, which plays the role of modeling and simulation technique for complex systems, such as spatial objects on a map. The TOPSIS method gives an approach to the geometric calculation between the distances of each alternative with the ideal alternative. Or other methods that implement different scientific concepts in decision-making. While these methods have grown in reputation over the years, the AHP method is still widely accepted by the community, due to its solid theoretical foundations and manageable simplicity. In the same way, the objective of this study had an approach aimed at evaluating the areas of high groundwater potential, opting for the use of a simple, reliable, and flexible method in the execution of the research.

Reviewer 3 Report

Thanks to the editor for the invitation to review the paper. It is about AHP and GIS application for evaluation the potential of groundwater. Comments
a) Abstract must be more precise with focus to objective, why the study is needed?
b) Brief results must be shown in Abstract and at present it is missing.
c) Why you have implemented AHP for determining criteria weights? This should be clearly explained? Why not LBWA, BWM or DIBR methods? These methods must be discussed and compared with AHP.
d) What are the challenges in earlier studies is not clear. Give examples to convince readers of the claim.
e) Some contributions are given, but its impact and insight is missing at present.
f) Literature review is critical, please improve on this, state clearly what is the focus of the paper and why the focus is presented.
What importance it brings to study line is not clear. For one aspect, AHP is essential in MCDM, but is there any specific focus is not recognized in the present work. Some important references with AHP application are missing, like below references: Sivaprakasam, P., & Angamuthu, M. (2023). Generalized Z-fuzzy soft β-covering based rough matrices and its application to magdm problem based on AHP method. Decision Making: Applications in Management and Engineering. https://doi.org/10.31181/dmame04012023p; Bakır, M., & Atalık, Özlem. (2021). Application of Fuzzy AHP and Fuzzy MARCOS Approach for the Evaluation of E-Service Quality in the Airline Industry. Decision Making: Applications in Management and Engineering, 4(1), 127-152.
g) Many symbols are there, please explain all clearly to help readers understand.
h) Properties with proof is fine, but what is the add-on it brings to the field is essential, please focus on that.
i) Comparison can be made effective too with more results and discussion and also expand the experimental setup

Author Response

We appreciate the review of the reviewer, as it allowed us to improve the paper

  1. a) Abstract must be more precise with focus to objective, why the study is needed?
    b) Brief results must be shown in Abstract and at present it is missing.

Answer:

We added the following, line 21:

The main objective of this research is to determine potential sources of groundwater using a Multi-Criteria Decision-Making technique with remote sensors. This article provides a method of exploration using Analytical Hierarchy Process (AHP) techniques applied to remote sensing data. The AHP method allows calculating the influence and along with the GIS environment, a map of groundwater exploitation potential can be produced. The results of GWPZs showed four classifications of groundwater potential. The distribution shows 15.02%, 23.93%, 59.80%, and 1.25% of the total area with high, moderate, low, and very low potential, respectively.

  1. c) Why you have implemented AHP for determining criteria weights? This should be clearly explained? Why not LBWA, BWM or DIBR methods? These methods must be discussed and compared with AHP.

Answer:

The following paragraph was added at the end of the discussion. Line 468

In recent years, new alternative methodologies have been developed from the classic concept applied in the AHP. Among these is Fuzzy AHP, which plays the role of modeling and simulation technique for complex systems, such as spatial objects on a map. The TOPSIS method gives an approach to the geometric calculation between the distances of each alternative with the ideal alternative. Or other methods that implement different scientific concepts in decision-making. While these methods have grown in reputation over the years, the AHP method is still widely accepted by the community, due to its solid theoretical foundations and manageable simplicity. In the same way, the objective of this study had an approach aimed at evaluating the areas of high groundwater potential, opting for the use of a simple, reliable, and flexible method in the execution of the research.

  1. d) What are the challenges in earlier studies is not clear. Give examples to convince readers of the claim.

Answer.

A paragraph was added in line 451.

There is currently much controversy in quantifying the recharge of the Caplina aquifer, this leads to studies to identify areas of high groundwater potential. Thus, this proposed model will be very useful to update and calibrate existing hydrogeological models in the Caplina Aquifer.

  1. e) Some contributions are given, but its impact and insight is missing at present.

Answer:

We demonstrated that the AHP technique combined with remote sensing in a GIS can predict potential groundwater in arid regions as in the Caplina Aquifer. On the other hand, we show the parameters that have the greatest influence in determining the potential of groundwater in this part of the world. This is aligned with our research questions which are discussed in the results and discussion and consolidated in the conclusions.

  1. f) Literature review is critical, please improve on this, state clearly what is the focus of the paper and why the focus is presented.

What importance it brings to study line is not clear. For one aspect, AHP is essential in MCDM, but is there any specific focus is not recognized in the present work. Some important references with AHP application are missing, like below references: Sivaprakasam, P., & Angamuthu, M. (2023). Generalized Z-fuzzy soft β-covering based rough matrices and its application to magdm problem based on AHP method. Decision Making: Applications in Management and Engineering. https://doi.org/10.31181/dmame04012023p; Bakır, M., & Atalık, Özlem. (2021). Application of Fuzzy AHP and Fuzzy MARCOS Approach for the Evaluation of E-Service Quality in the Airline Industry. Decision Making: Applications in Management and Engineering, 4(1), 127-152.

Answer:

We incorporated the suggested references, 73 and 74.

Round 2

Reviewer 3 Report

All the reviewers' commrnts have been adressed carefully and sufficiently, the revisions are rational from my point of view. I think the current version of the paper can be accepted.